# Intra-industry peer effect in corporate environmental information disclosure: Evidence from China

**Kewei Hu**[1], **Yugui Hao**[1]*, **Dan Yu**[2]

1 College of Economics and Management, Zhejiang Agriculture and Forestry University, Hangzhou, Zhejiang, China, 2 College of Continuing Education, Zhejiang Agriculture and Forestry University, Hangzhou, Zhejiang, China

* yghao@zafu.edu.cn

**Data Availability Statement:** All relevant data are within the paper and its Supporting Information files.

## Abstract

Corporate environmental information disclosure is an important way for stakeholders to understand the performance of corporate environmental responsibilities. To explore the group relevance of corporate environmental information disclosure, this paper empirically tests the intra-industry peer effect of corporate environmental information disclosure using a panel fixed-effects model based on data of Chinese heavily polluted listed companies from 2015 to 2019 and studies its formation mechanism and impact path. The results show that there is an intra-industry peer effect in corporate environmental information disclosure; this effect exists in corporations of different ownership; social learning mechanism and environmental pressure mechanism are the channels to form the intra-industry peer effect of corporate environmental information disclosure; there are both intra-group imitation and inter-group imitation in the intra-industry peer effect of corporate environmental information disclosure. Based on the research results, the government can select corporations in various industries with excellent quality of environmental information disclosure as benchmarks to provide learning templates for corporations with inferior information. At the same time, the government can impose appropriate environmental protection pressure to promote learning and imitation among corporations. It is important to note that when selecting benchmarking companies, priority should be given to large and high-performing corporations.

## 1. Introduction

With the rapid development of the Chinese economy, the public has gradually realized that the increasingly severe environmental problems will restrict economic development and endanger human survival. Balancing the relationship between economic growth and natural resource utilization to achieve harmonious development of industry and ecosystem has become the focus [1, 2]. Chinese leaders have repeatedly emphasized, "The construction of ecological civilization is the fundamental plan for the sustainable development of the Chinese nation." Green development and ecological civilization construction have been elevated to an unprecedented

**Funding:** The research was funded by the National Social Science Fund of China, Approval No.16BGL070.

**Competing interests:** The authors have declared that no competing interests exist.

strategic level. The studies showed that the industrial and agricultural sectors are the main sources of environmental pollution and climate change [3–6]. Understanding the adoption of measures is necessary to mitigate climate change and excessive fossil fuel use [7, 8]. At the same time, corporations should expand the scope of entrusted responsibilities and assume responsibility for environmental management and protection. Investors, governments, the public, and other stakeholders are also paying more attention to the environmental performance of corporations. As an essential part of the corporate environmental management system, environmental information disclosure can effectively meet these urgent needs.

Corporate environmental information disclosure (EID) refers to the corporate disclosure of information related to the natural environment during its operation, which is the embodiment of corporate fulfillment of reporting responsibility in the principal-agent relationship. Corporate EID meets the needs of external stakeholders for corporate environmental information, which can directly or indirectly affect corporate value and investors' decision-making [9]. At the same time, EID is one of the tools for the government to manage public goods. The increasingly severe environmental degradation forces the government to continuously increase environmental protection requirements for corporations. On May 24, 2021, the Ecological and Environmental office of the Chinese State Council issued *"the Reform Plan for the Legal Disclosure of Environmental Information"*, clearly stating that a mandatory disclosure system for corporate environmental information should be formed by 2025.

Existing literature studies the influencing factors of corporate EID from both external and internal perspectives. The external perspective mainly examines the impact of environmental regulation [10–14], regional economic development level [15], market pressure [16], and media attention [17–20] on corporate EID. The research from the internal perspective can be further divided into corporate governance perspective and environmental protection behavior perspective. The corporate governance perspective studies how corporate characteristics affect the level of EID, such as management capabilities [21], company size [22–25], corporate profitability [26], and the proportion of independent directors [27]. From the perspective of environmental protection behavior, more attention is paid to the impact of environmental protection investment [28] and environmental performance [29–35] on the level of EID.

In summary, there is a large volume of published studies analyzing the influence factors of corporate EID. The theoretical results provided an important reference for the government to formulate relevant policies to improve the level of corporate EID. However, most studies have only focused on the influence of corporate factors or common external factors on EID. Few writers have been able to draw on any systematic research into the mutual influence between corporations in EID. Recently, researchers have shown an increased interest in the imitation behavior of corporations on CSR disclosure and EID [36, 37]. Along this line of research, several questions need to be considered in depth. Is there a convergence in the quality of EID among companies in the same region, in the same industry, or with other links to each other? Why does this peer effect arise? How can this peer effect be used to improve the overall level of corporate EID? Based on the above considerations, this paper examines the peer effect of corporate EID from the perspective of the same industry and explores its formation mechanism and influence path. The following are innovations and contributions of this paper: (1) This paper studies the group correlation of corporate EID in the same industry and examines whether this phenomenon exists in corporations with different ownerships, which enriches the theoretical basis of research on EID; (2) In addition to testing the existence of the intra-industry peer effect in the corporate EID, this paper also empirically tests the formation mechanism of the peer effect, explores its influence path based on the law of imitation, enriching the research on peer effect and providing a reference for research on peer effect in other corporate governance decisions.

## 2. Research hypotheses

### 2.1 The intra-industry peer effect in corporate environmental information disclosure

The Peer effect, which originated in sociology, refers to the phenomenon that an individual's behavior is affected by the group's behavior to a certain extent and changes with the change of the group's behavior [38]. The ancient Chinese saying "what's near cinnabar goes red, and what's next to ink turns black" is the embodiment of this phenomenon. The earliest research on the peer effect was mainly concentrated in sociology, such as financial decisions of individuals and families [39] and criminal behavior of citizens [40]. Due to its prevalence, the peer effect has become a hot topic in sociology, education, economics, finance, and management. The decisions of corporations that are interconnected or in the same environment often show a convergence [41]. A firm's financial and operational decisions depend not only on the environment in which the firm operates but also on the behavior of other firms [42]. Existing studies of corporate peer effect have found that capital structure [43, 44], investment decisions [45], cash holdings [46], M&A decisions [47], executive compensation [48], dividend policy [49], innovation behavior [50] and other financial and corporate governance decisions have noticeable peer effects. In terms of social responsibility, Peng [51] found that corporate charitable donations will be significantly affected by the charitable donations of peer corporations; Wen [52] studied the peer effect of corporate poverty alleviation behavior and found that the investment in poverty alleviation of other corporations in the same industry will significantly and positively affect corporate poverty alleviation decisions. Like charitable donations and poverty alleviation, EID is also a part of corporate social responsibility. Thus, corporate EID is likely to have an intra-industry peer effect.

At the same time, considering the significant differences between Chinese state-holding corporations and non-state-holding corporations in terms of liability burden, corporate governance, and role in the market [53], it is necessary to further verify the intra-industry peer effect in EID of corporations with different ownerships. Compared with non-state-holding corporations, the chairman and senior managers in most Chinese state-holding corporations are appointed by the government. This means that non-state-holding corporations are required to work on achieving political goals in addition to economic goals [54]. Due to their special political status, state-owned corporations naturally pay more attention to the fulfillment of social responsibility, leading to the fact that they may show "competitiveness" rather than "fellowship" in EID. In addition to universal regulations such as *"the Guidelines on Environmental Information Disclosure for Listed Companies"* and *"the Measures for the Legal Administration of Corporate Environmental Information Disclosure"*, state-holding corporations face additional environmental regulatory pressure imposed by the Chinese State-owned Assets Supervision and Administration Commission. When making environmental disclosures, state-holding corporations may be more inclined to comply with the authorities' requirements and less sensitive to their peers' behavior. Therefore, this paper proposes the following assumptions.

**Hypothesis 1a.** *There is an intra-industry peer effect in corporate environmental information disclosure.*

**Hypothesis 2a.** *There is an intra-industry peer effect in the environmental information disclosure of state-controlled corporations.*

**Hypothesis 3a.** *There is an intra-industry peer effect in the environmental information disclosure of non-state-controlled corporations.*

## 2.2 Formation mechanism of intra-industry peer effect in corporate environmental information disclosure

Scholars have pointed out that the influence mechanism of imitation behavior is mainly divided into two types. One is based on information theory, and the other is on competition theory [55]. The former believes that companies imitate the behavior of peer companies with information superiority to obtain valuable decision-making information. The latter assumes that companies imitate the behavior of peer companies to maintain their relative position or suppress competitors. Considering the current situation of Chinese corporate EID, the above two mechanisms can be divided into social learning and environmental pressure mechanisms.

**2.2.1 Social learning mechanism.** As a kind of corporate information disclosure, the essence of EID is to realize the corporate value by improving the information asymmetry between the corporations and the stakeholders. Currently, most of the existing corporate EID is voluntary disclosure rather than mandatory disclosure. Since there is no uniform standard, companies need to decide what to disclose and how to disclose it. Information-based imitation theory argues that in an environment of uncertainty and ambiguity, the decisions of corporations in the same industry may be an essential source of information. Research also points out that if some people or corporations are perceived as likely to have more information, they can become "Fashion Leaders" [56]. Studies have also found that larger, more competitive organizations are more likely to be imitated [57]. Therefore, in this uncertain environment, the EID of peer companies has become an essential source of information for companies to conduct environmental disclosure. By imitating the EID of peer companies, the companies with information disadvantages can reduce the cost of information search and make relatively reasonable environmental disclosures. In summary, this paper proposes the following assumptions.

> **Hypothesis 2a.** *Corporations with information disadvantages in the market are more susceptible to the influence of peer effects in their environmental information disclosure.*

**2.2.2 Environmental protection pressure mechanism.** To adapt to the system and culture of the society where the corporation operates, the corporation must meet the local community's expectations for its behavior. If the local government management agency makes an explicit request, or if the local peer companies produce high-quality EID, the corporation will feel the pressure of the public and will disclose according to the requirements or imitate other companies. In other words, when an organization is under particular pressure, a safe way to adjust its behavior is to mimic the behavior of other recognized organizations. Scholars have found that the disclosure level of American companies is significantly higher than that of Canadian companies in terms of corporate compliance with environmental regulations. This is because American society prefers legal proceedings to resolve conflicts, leading companies to disclose relevant environmental information as much as possible to avoid investor lawsuits [58]. Chinese current system has created institutional pressure on the EID of listed companies, which has significantly affected the EID behavior of listed companies [59]. In summary, to maintain their competitive advantages, companies generally succumb to environmental pressure and actively imitate the EID of peer companies. Based on this, this paper proposes the following assumptions.

> **Hypothesis 2b.** *Corporations that face more significant environmental protection pressure in the market are more susceptible to the influence of peer effects in their environmental information disclosure.*

## 3. Research design

### 3.1 Sample selection and data sources

Since the EID of companies in heavily polluting industries is representative, this paper selects Chinese A-share listed companies in heavily polluting industries from 2015 to 2019 as the research object. After excluding the missing data and the samples of ST and PT listed companies, 5-year data of 477 sample companies were obtained, totaling 2375 sample observations. Among the variable data used in this paper, the level of EID is obtained by reading the annual reports, social responsibility reports, and environmental reports of listed companies. Other variables are obtained from the CSMAR database and the National Bureau of Statistics official website. All continuous variables were winsorized at the 1% level to eliminate the influence of extreme values.

### 3.2 Variable description

**3.2.1 Measurement of environmental information disclosure index.** This paper uses the indicator evaluation method to measure the level of corporate EID. Referring to Wu's [60] indicator, this paper divides environmental information into monetized and non-monetized information. The monetized environmental information mainly comes from financial reports, financial statement notes, and supplementary statements. The non-monetized environmental information mainly comes from the rest of the annual report, social responsibility report, sustainable development report, and environmental report. There are six monetized environmental information indicators and seven non-monetized environmental information indicators. Considering that quantitative data is more reliable than qualitative data, 2 points are assigned to indicators that combine quantitative and qualitative data, 1 point is assigned to only qualitative indicators, and 0 is assigned to undisclosed indicators. The final level of corporate EID is the sum of the scores of each project. The optimal score is 26 points, and the specific composition of the indicators is shown in Table 1.

**3.2.2 Moderating variable.** To verify the social learning mechanism and environmental protection pressure mechanism of intra-industry peer effect in corporate EID, this paper

**Table 1. Measures of EID indicators.**

| Category | Indicator | Numerical Value | | |
|---|---|---|---|---|
| | | Undisclosed | Qualitative Disclosure | Qualitative And Quantitative Disclosure |
| Monetized Indicators | Sewage Charges and Environmental Tax | 0 | 1 | 2 |
| | Emergency Expenses for Major Environmental Issues | 0 | 1 | 2 |
| | Environmental Investment Spending or Borrowing | 0 | 1 | 2 |
| | Benefits of Reduce Pollution | 0 | 1 | 2 |
| | Income from Waste Utilization | 0 | 1 | 2 |
| | Environmental Grant Relief and Incentive Income | 0 | 1 | 2 |
| Non-monetized Indicators | Environmental Information Disclosure System | 0 | 1 | 2 |
| | Environmental Management Goals | 0 | 1 | 2 |
| | Environmental Measures and Improvements | 0 | 1 | 2 |
| | Perform Certification | 0 | 1 | 2 |
| | Energy Saving Measures and Results | 0 | 1 | 2 |
| | Types of Pollution Discharges and Emission Compliance | 0 | 1 | 2 |
| | Independent Social Responsibility Sustainability Report or Independent Environmental Report | Have for 2, Not have for 0 | | |

introduces information advantage and environmental protection pressure as moderator variables. Corporations that have entered the capital market earlier have made more environmental disclosures. These older companies have accumulated more experience in EID and are often considered to have mature disclosure content and paradigms that meet the expectations of investors and regulators. Similar to other peer effect studies [51], this paper selects the listing year as a proxy variable for information advantage. If the year of the listing is more than five years, the corporation is considered "experienced" among peer corporations and has an information advantage in making EID decisions. Otherwise, the corporation is deemed to be in a disadvantaged information position. When the corporation is in the information advantage, the variable takes the value of 1. Otherwise, it takes the value of 0. The measurement of environmental protection pressure adopts the text analysis method to count the ratio of the number of sentences containing environmental protection words to the total number of sentences in the government work report of the province where the corporation is registered. This paper uses this ratio to measure environmental protection pressure because the content of local government reports on the environment is a concentrated expression of the government's environmental awareness, which is an essential factor in determining the environmental pressure of corporations. The larger the ratio, the greater the environmental protection pressure corporations face.

**3.2.3 Control variables.** To accurately measure the impact of intra-industry peer effect on the level of corporate EID, this paper introduces company size, equity concentration, asset-liability ratio, return on assets, Tobin q, operating income growth rate, auditor size, and audit opinion as control variables. In addition, since corporations will follow the disclosure content and paradigm of the previous EID to a certain extent, this paper introduces the EID level of the previous period to control the impact of the previous disclosure habits on the disclosure behavior in the current period.

The variables for this study are summarized in Table 2.

**Table 2. Summary table of variables.**

| Variable Type | Variable Name | Variable Symbol | Variable Definitions |
|---|---|---|---|
| Explained Variable | EDI Level | EDI | Measure EID score according to Table 1 |
| Explanatory Variable | Industry EDI level | MARKET | The average value of the disclosed scores of other companies in the industry |
| Moderating Variable | Environmental Protection Pressure | PRESS | The ratio of environmental protection sentences in the government report of the province where the company registered |
| | Listing Years | AGE | If the company has been listed for less than five years, the value is 0. Otherwise, the value is 1. |
| Control Variable | The level of EID in the previous period | $EDI_{t-1}$ | EID score in the last period |
| | Company Size | SIZE | Natural logarithm of total assets at the end of the year |
| | Equity Concentration | CR | The total shareholding ratio of the top 10 shareholders |
| | Asset-liability Ratio | LEV | Total liabilities at the end of the year/Total assets at the end of the year |
| | Return on Assets | ROA | Net profit/Total assets at the end of the year |
| | Tobin Q | Q | Company market value/Asset replacement cost |
| | Operating Income Growth Rate | GROWTH | Operating income growth in the current period/Operating income in the previous period |
| | Auditor Size | BIG4 | If the company's audit firm is Big 4, the value is 1. Otherwise, the value is 0. |
| | Audit Opinion | OPIN | If the company has been given a non-standard audit opinion on its annual financial statements, the value is 1. Otherwise, the value is 0. |

### 3.3 Model specification

Considering the characteristics of the data, to test hypotheses 1a, 1b, and 1c, this paper adopts the robust fixed-effect model regression method to control the individual fixed effects of all companies. Based on the related studies on the peer effect, the following model is constructed in this paper [29].

$$EDI_{i,t} = \alpha + \beta MARKET_{i,t-1} + \lambda \sum CONTROLS + \varepsilon_{i,t} \tag{1}$$

In Formula (1), $EDI_{i,t}$ represents the EID level of the company i in the current period; $MARKET_{i,t-1}$ represents the average level of EID in the industry of company i after excluding company i in the previous period; CONTROLS is a control variable; $\varepsilon_{i,t}$ represents the random disturbance term which is assumed to be normality distributed with zero mean value and constant variance [61]. When verifying hypotheses 1a, 1b, and 1c, regression was performed by substituting samples of all companies, state-owned holding companies, and non-state-owned holding companies, respectively. Since β in the model measures the influence of peer effects, we will focus on whether it is significant.

To test hypothesis 2a, this paper constructed the following model based on Eq (1).

$$EDI_{i,t} = \alpha + \beta_1 MARKET_{i,t-1} + \beta_2 AGE_{i,t} + \beta_3 MARKET_{i,t-1} \times AGE_{i,t} + \lambda \sum CONTROLS \\ + \varepsilon_{i,t} \tag{2}$$

Based on Eq (1), Eq (2) adds the company's listing year $AGE_{i,t}$ and the interaction term $MARKET_{i,t-1} \times AGE_{i,t}$ between the listing year and the average EID level of industry in the previous period. Because $\beta_3$ in the model measures the moderating effect of the social learning mechanism, we will focus on whether this coefficient is significant.

To test hypothesis 2b, this paper constructed the following model based on Eq (1).

$$EDI_{i,t} = \alpha + \beta_1 MARKET_{i,t-1} + \beta_2 PRESS_{i,t} + \beta_3 MARKET_{i,t-1} \times PRESS_{i,t} + \lambda \sum CONTROLS \\ + \varepsilon_{i,t} \tag{3}$$

Based on Eq (1), Eq (3) adds the environmental protection pressure $PRESS_{i,t}$, and the interaction term $MARKET_{i,t-1} \times PRESS_{i,t}$ between the environmental protection pressure and the average EID level of industry in the previous period. Because β3 in the model measures the moderating effect of the environmental pressure mechanism, we will focus on whether this coefficient is significant. Since both variables that make up the interaction term are non-dummy variables, the variables $MARKET_{i,t-1}$ and $PRESS_{i,t}$ in the model are centralized to make the coefficients of $MARKET_{i,t-1}$ and $PRESS_{i,t}$ themselves more meaningful and comparable [62].

## 4. Results and discussion

### 4.1 Descriptive statistics and correlation analysis

Table 3 shows the descriptive statistical results of the main variables. Among them, the minimum value, maximum value, and mean value of Edi are 0, 24, and 10.567, respectively, indicating a massive gap between the best-performing corporations and the worst-performing corporations in terms of EID and the overall EID level of sample companies is low. The mean value of Market is 10.107, and the standard deviation is 2.678, indicating that there is little difference in the level of EID among different industries. The mean value of Big4 is only 0.075, indicating that only 7.5% of the companies in the sample choose the Big Four accounting firms for auditing. The mean value of Opin is 0.02, suggesting that most corporate financial

**Table 3. Descriptive statistical results of variables.**

| Variable | Mean | Standard Deviation | Minimum | Maximum |
|---|---|---|---|---|
| Edi | 10.567 | 4.948 | 0 | 24 |
| Market | 10.572 | 2.673 | 2 | 17.455 |
| Size | 22.594 | 1.325 | 20.212 | 26.322 |
| Cr | 35.229 | 14.57 | 9.442 | 74.566 |
| Lev | 0.399 | 0.193 | 0.059 | 0.84 |
| Roa | 0.046 | 0.057 | -0.148 | 0.213 |
| Tobin Q | 2.136 | 1.409 | 0.827 | 8.357 |
| Growth | 0.199 | 0.587 | -0.701 | 4.191 |
| Big4 | 0.075 | 0.263 | 0 | 1 |
| Opin | 0.02 | 0.141 | 0 | 1 |

reports have obtained standard audit opinions in this time range. The mean value of Roa was only 0.046, indicating that the sample companies' profitability was low from 2015 to 2019. Lev and other financial indicators are in the normal range, which is consistent with the findings of other studies.

Table 4 shows the correlation analysis results of the main variables. It can be seen from Table 4 that Edi is significantly positively correlated with Market (R = 0.493, P < 0.01). Meanwhile, Edi is significantly positively or negatively correlated with control variables such as Size, Cr, Lev, Roa, Tobin Q, Growth, Big4, and Opin, indicating that it is appropriate to control these variables. Except that the correlation coefficient between Edi and Size is higher than 0.5, the absolute values of correlation coefficients between other variables are all low, indicating that the regression model does not have serious multicollinearity problems.

## 4.2 Regression analysis

**4.2.1 The existence test of intra-industry peer effect in corporate EID.** Table 5 lists the regression results of the total sample corporations and the grouped regression results considering the nature of ownership. Column (1) lists the regression results of the total sample corporations. The regression coefficient of $Market_{t-1}$ is 0.259, which is significant at the 1% level,

**Table 4. Correlation analysis.**

| Variables | Edi | Market | Size | Cr | Lev | Roa | Tobin Q | Growth | Big4 | Opin |
|---|---|---|---|---|---|---|---|---|---|---|
| Edi | 1.000 | | | | | | | | | |
| Market | 0.493*** | 1.000 | | | | | | | | |
| Size | 0.529*** | 0.376*** | 1.000 | | | | | | | |
| Cr | 0.165*** | 0.151*** | 0.329*** | 1.000 | | | | | | |
| Lev | 0.300*** | 0.295*** | 0.497*** | 0.093*** | 1.000 | | | | | |
| Roa | -0.051** | -0.112*** | -0.049** | 0.083*** | -0.435*** | 1.000 | | | | |
| Tobin Q | -0.382*** | -0.383*** | -0.464*** | -0.100*** | -0.352*** | 0.300*** | 1.000 | | | |
| Growth | -0.126*** | -0.051** | -0.117*** | -0.056*** | 0.032 | -0.042** | 0.013 | 1.000 | | |
| Big4 | 0.185*** | 0.048** | 0.365*** | 0.153*** | 0.082*** | 0.032 | -0.116*** | -0.055*** | 1.000 | |
| Opin | -0.035* | 0.030 | -0.062*** | -0.073*** | 0.096*** | -0.171*** | 0.045** | 0.014 | -0.041** | 1.000 |

*** p<0.01,

** p<0.05,

* p<0.1.

**Table 5. Regression test results of intra-industry peer group effect in EID.**

| VARIABLES | (1) | (2) | (3) |
|---|---|---|---|
| | All corporations | State-holding corporations | Non-state-holding corporations |
| Market$_{t-1}$ | 0.259*** | 0.304*** | 0.254*** |
| | (6.79) | (4.92) | (5.21) |
| Edi$_{t-1}$ | 0.337*** | 0.248*** | 0.382*** |
| | (12.20) | (5.31) | (11.69) |
| Size | 1.585*** | 1.775*** | 1.309*** |
| | (5.28) | (4.74) | (2.75) |
| Cr | -0.022 | -0.036* | 0.000 |
| | (-1.62) | (-1.87) | (0.01) |
| Lev | 0.560 | -0.966 | 1.362 |
| | (0.67) | (-0.61) | (1.36) |
| Roa | 0.891 | -1.289 | 1.973 |
| | (0.62) | (-0.47) | (1.08) |
| Tobin Q | -0.451*** | -0.603*** | -0.440*** |
| | (-5.18) | (-4.16) | (-4.02) |
| Growth | -0.046 | 0.177 | -0.230* |
| | (-0.40) | (1.03) | (-1.89) |
| Big4 | -1.807** | -2.476* | -1.369 |
| | (-2.40) | (-1.93) | (-1.58) |
| Opin | -0.072 | 0.439 | -0.159 |
| | (-0.15) | (0.47) | (-0.29) |
| Constant | -29.174*** | -31.533*** | -24.323** |
| | (-4.45) | (-3.69) | (-2.32) |
| Observations | 1,848 | 778 | 1,042 |
| R-squared | 0.489 | 0.451 | 0.522 |
| Company FE | YES | YES | YES |

Robust t-statistics in parentheses

*** p<0.01,

** p<0.05,

* p<0.1.

indicating an intra-industry peer effect in corporate EID. Hypothesis 1a is proved, and this finding is consistent with that of Shen and Su, who verified the mutual influence of corporate EID [37]. Columns (2) and (3) show the regression results of state-owned and non-state-owned holding corporations. One unanticipated finding was that the regression coefficient of Market $_{t-1}$ in Columns (2) and (3) are all significant at the 1% level, indicating that both state-holding corporations and non-state-holding corporations will be affected by peer effect when they disclose environmental information, supporting research hypotheses 1b and 1c. At the same time, in all regression tests, the regression coefficients of Edi$_{t-1}$ were significantly positive at the level of 1%, indicating that corporate EID behavior in the current period would be affected by the corporate EID level in the previous period, which accords with previous studies [37].

**4.2.2 Examination of the formation mechanism of intra-industry peer effect in corporate EID.** The test results of hypothesis 2 and hypothesis 3 are presented in Table 6. Column (1) lists the regression test results of the social learning mechanism. The regression coefficient of Age×Market$_{t-1}$, which is the interaction term between listing years and the EID level of

**Table 6. Regression test results of intra-industry peer group effect in EID.**

| VARIABLES | (1) | (2) |
|---|---|---|
| | Social learning mechanism | Environmental protection pressure mechanism |
| Market$_{t-1}$ | 0.511*** | 0.280*** |
| | (4.59) | (7.14) |
| Age×Market$_{t-1}$ | -0.252** | |
| | (-2.29) | |
| Age | 1.693* | |
| | (1.73) | |
| Press×Market$_{t-1}$ | | 3.942*** |
| | | (3.98) |
| Press | | 5.878*** |
| | | (1.71) |
| Edi$_{t-1}$ | 0.340*** | 0.336*** |
| | (12.22) | (12.27) |
| Size | 1.578*** | 1.554*** |
| | (5.37) | (5.22) |
| Cr | -0.021 | -0.019 |
| | (-1.52) | (-1.41) |
| Lev | 0.334 | 0.565 |
| | (0.40) | (0.69) |
| Roa | 0.619 | 0.776 |
| | (0.44) | (0.55) |
| Tobin Q | -0.428*** | -0.438*** |
| | (-4.86) | (-5.03) |
| Growth | -0.062 | -0.064 |
| | (-0.52) | (-0.56) |
| Big4 | -1.754** | -1.824** |
| | (-2.24) | (-2.43) |
| Opin | -0.035 | -0.052 |
| | (-0.07) | (-0.11) |
| Constant | -30.751*** | -25.912*** |
| | (-4.76) | (-3.94) |
| Observations | 1,848 | 1,848 |
| R-squared | 0.492 | 0.494 |
| Company FE | YES | YES |

Robust t-statistics in parentheses

***p<0.01,

** p<0.05,

* p<0.1

corporations in the same group, is -0.252 and passes the significance test of 5%, showing that the earlier a corporation enters the capital market, the less it is affected by the peer effect. This implies that younger corporations are more susceptible to peer effects, which is in line with previous research findings [51, 63]. The industrial peer effect of EID will be weakened by corporate information advantage. Corporations with inferior information in the market are more likely to be affected by peer effects in EID. Thus, hypothesis 2 has been verified. This indicates that the intra-industry peer effect in corporate EID is partly due to the social learning mechanism. Column (2) lists the regression test results of the environmental protection pressure

mechanism. The regression coefficient of Press×Market$_{t-1}$, the interaction term between environmental protection pressure and the EID level of corporations in the same group, is 3.909 and passes the significance test of 1%. This indicates that environmental protection pressure has a positive moderating effect on the industrial peer effect of corporate EID. The greater the pressure of environmental protection faced by corporations, the more likely their EID will be affected by peer effect. Hypothesis 3 has been verified. This indicates that the peer effect of EID is partly due to the environmental protection pressure mechanism.

### 4.3 Robustness test

Referring to the practice of Shen and Su [37], this paper uses the relative level of EID instead of the original absolute level of EID as the explained variable to conduct regression tests. The specific method takes the ratio of the actual corporate score and the maximum possible score as the EID index, representing the relative EID level. As shown in Table 7, the regression coefficients of the average corporate EID level (Market$_{t-1}$) in the first three columns are 0.264, 0.293, and 0.258, respectively, which are significant at the 1% level. In column (4), the regression coefficient of the interaction term (Age×Market$_{t-1}$) between the listing years of corporations and the EID level of peer corporations is -0.291, which is significant at the 5% level. In column (5), the regression coefficient of the interaction term (Press×Market$_{t-1}$) between environmental protection pressure and the EID level of peer corporations is 3.957, which is significant at a 1% level. The empirical results have not changed substantially, proving that the research conclusions are robust and reliable.

## 5. Further study

To further explore the influence path of peer effect in corporate EID, this paper analyzes the imitation law of corporate EID behavior from the perspective of corporate size and corporate governance. According to corporation size (total assets) and corporation performance (return on total assets), the total sample is divided into small and large companies, low-performance and high-performance corporations.

### 5.1 Corporation size factor

This paper focuses on the corporation size factor because many previous studies have found that corporation size can affect the degree of imitation and peer effect. For example, Leary [42] argues that there is a peer effect in corporate financing decisions, in which small firms are more likely to imitate their larger peers. To test the imitation law of corporate EID in terms of corporation size, this paper divides corporations into groups based on the index of total assets. The top 50% of corporations in terms of assets are large corporations, and the bottom 50% are small corporations. Further study continues to use the fixed effects model for regression. Based on model (1), the industry Edi average item is further split into the large corporation Edi average item and the small corporation Edi average item, thus forming the following new model.

$$EDI_{i,t} = \alpha + \beta_1 Baedi_{i,t-1} + \beta_2 Saedi_{i,t-1} + \lambda \sum CONTROLS + \varepsilon_{i,t} \tag{4}$$

In the model, Baedi$_{t-1}$ and Saedi$_{t-1}$ represent the average disclosure values of large corporations and small corporations in the last period after the elimination of corporation i.

To test whether small corporations imitate large corporations and whether there is mutual imitation among small corporations, this paper takes small corporations as samples to perform regression on the model (4). If the regression coefficient $\beta_1$ is significantly positive, it indicates

**Table 7. Robustness test results.**

| VARIABLES | (1) | (2) | (3) | (4) | (5) |
|---|---|---|---|---|---|
| | All corporations | State-holding corporations | Non-state-holding corporations | Social learning mechanism | Environmental protection pressure mechanism |
| Market$_{t-1}$ | 0.259*** | 0.304*** | 0.254*** | 0.511*** | 0.281*** |
| | (6.79) | (4.92) | (5.21) | (4.59) | (5.66) |
| Age×Market$_{t-1}$ | | | | -0.252** | |
| | | | | (-2.29) | |
| Age | | | | 0.071* | |
| | | | | (1.73) | |
| Press×Market$_{t-1}$ | | | | | 5.853*** |
| | | | | | (4.05) |
| Press | | | | | 0.433** |
| | | | | | (2.16) |
| Edi$_{t-1}$ | 0.337*** | 0.248*** | 0.382*** | 0.340*** | 0.382*** |
| | (12.20) | (5.31) | (11.69) | (12.22) | (11.84) |
| Size | 0.066*** | 0.074*** | 0.055*** | 0.066*** | 0.052*** |
| | (5.28) | (4.74) | (2.75) | (5.37) | (2.71) |
| Cr | -0.001 | -0.001* | 0.000 | -0.001 | -0.000 |
| | (-1.62) | (-1.87) | (0.01) | (-1.52) | (-0.06) |
| Lev | 0.023 | -0.040 | 0.057 | 0.014 | 0.061 |
| | (0.67) | (-0.61) | (1.36) | (0.40) | (1.48) |
| Roa | 0.037 | -0.054 | 0.082 | 0.026 | 0.081 |
| | (0.62) | (-0.47) | (1.08) | (0.44) | (1.07) |
| Tobin Q | -0.019*** | -0.025*** | -0.018*** | -0.018*** | -0.018*** |
| | (-5.18) | (-4.16) | (-4.02) | (-4.86) | (-3.88) |
| Growth | -0.002 | 0.007 | -0.010* | -0.003 | -0.009* |
| | (-0.40) | (1.03) | (-1.89) | (-0.52) | (-1.89) |
| Big4 | -0.075** | -0.103* | -0.057 | -0.073** | -0.060* |
| | (-2.40) | (-1.93) | (-1.58) | (-2.24) | (-1.70) |
| Opin | -0.003 | 0.018 | -0.007 | -0.001 | -0.006 |
| | (-0.02) | (0.47) | (-0.29) | (-0.07) | (-0.23) |
| Constant | -1.216*** | -1.314*** | -1.013*** | -1.281*** | -0.838** |
| | (-4.45) | (-3.69) | (-2.32) | (-4.76) | (-1.99) |
| Observations | 1,848 | 778 | 1,042 | 1,848 | 1,848 |
| R-squared | 0.489 | 0.451 | 0.522 | 0.492 | 0.531 |
| Company FE | YES | YES | YES | YES | YES |

Robust t-statistics in parentheses

***p<0.01,

** p<0.05,

* p<0.1.

that small corporations imitate large corporations. If the regression coefficient $\beta_2$ is significantly positive, it suggests that there is mutual imitation among small corporations. To test whether there is mutual imitation among large corporations in the same industry, the regression of model (4) takes large corporations as samples. If the regression coefficient $\beta_1$ is significantly positive, it indicates that there is mutual imitation among large corporations.

In Table 8, Column (1) shows the regression results of the sample of large corporations. The regression coefficient of Baedi is 0.228 and significant at the 5% level, indicating that there

**Table 8. Grouping regression results of corporation size factors.**

| VARIABLES | (1) | (2) |
|---|---|---|
| | Sample of large corporations | Sample of small corporations |
| Baedi | 0.228** | 0.065 |
| | (2.47) | (0.45) |
| Saedi | 0.038 | 0.186 |
| | (0.41) | (1.36) |
| Edi$_{t-1}$ | 0.320*** | 0.323*** |
| | (6.78) | (8.25) |
| Size | 1.725*** | 1.624*** |
| | (2.76) | (2.70) |
| Cr | -0.040* | -0.032 |
| | (-1.83) | (-1.05) |
| Lev | 0.984 | 0.895 |
| | (0.70) | (0.65) |
| Roa | 4.790* | -3.238 |
| | (1.74) | (-1.54) |
| Tobin Q | -0.499** | -0.517*** |
| | (-2.50) | (-5.14) |
| Growth | -0.130 | -0.017 |
| | (-0.89) | (-0.09) |
| Big4 | -2.419*** | -0.802 |
| | (-2.99) | (-0.90) |
| Opin | 0.205 | -0.715 |
| | (0.18) | (-0.93) |
| Constant | -31.949** | -28.633** |
| | (-2.24) | (-2.23) |
| Observations | 936 | 874 |
| R-squared | 0.463 | 0.490 |
| Company FE | YES | YES |

Robust t-statistics in parentheses

*** $p<0.01$,

** $p<0.05$,

* $p<0.1$.

is mutual imitation among large corporations in the same industry. Column (2) lists the regression results of the sample of small corporations, in which the regression coefficient of Baedi is 0.065 and that of Saedi is 0.186, both of them are insignificant, implying that small corporations do not have a significant tendency to imitate small corporations or large corporations. In sum, in terms of corporation size, the intra-industry peer effect of corporate EID is mainly derived from the mutual imitation among large-scale corporations.

## 5.2 Corporate governance factors

According to the law of logical imitation, corporations with better corporate governance in the same industry may be considered more exemplary in their EID, making them easier to be imitated. On the other hand, according to the insider-after-exterior law, corporations with poor corporate governance quality within the same industry are more likely to emulate and learn from each other. Corporations are divided according to the return on total assets to distinguish

the above conflicting theoretical expectations. The top 50% are high-performance corporations, and the bottom 50% are low-performance corporations. Based on model (1), the industry Edi average item is further split into the high-performance corporation Edi average item and the low-performance corporation Edi average item, thus forming the following model.

$$EDI_{i,t} = \alpha + \beta_1 Haedi_{i,t-1} + \beta_2 Laedi_{i,t-1} + \lambda \sum CONTROLS + \varepsilon_{i,t} \qquad (5)$$

In the model, $Haedi_{t-1}$ and $Laedi_{t-1}$ respectively represent the average disclosure values of high-performance corporations and low-performance corporations in the last period after the elimination of corporation i.

To test whether low-performance corporations in the same industry imitate high-performance corporations and whether there is mutual imitation among low-performance corporations, low-performance corporations are taken as samples to conduct regression for the model (5). If the regression coefficient $\beta_1$ is significantly positive, it indicates that low-performance corporations imitate high-performance corporations. If the regression coefficient $\beta_2$ is significantly positive, it indicates that there is mutual imitation among low-performance corporations. To test whether there is mutual imitation among high-performing corporations in the same industry, high-performing corporations are taken as samples for model regression (5). If the regression coefficient $\beta_1$ is significantly positive, it indicates that there is mutual imitation among high-performing corporations.

In Table 9, Column (1) lists the regression results of high-performance corporations. The regression coefficient of haedi is 0.178 and significant at the 1% level, while that of laedi is 0.002 and not significant, indicating that high-performance corporations imitate each other and high-performance corporations do not imitate low-performance corporations. Column (2) lists the regression results of low-performance corporations. It is found that the regression coefficient of haedi is 0.272 and significant at the 1% level, while the regression coefficient of laedi is -0.013, which is not significant, indicating that low-performance corporations tend to imitate high-performance corporations rather than low-performance corporations. In sum, the above results show that the effect of corporate governance on peer effect is more in line with the expectation of the law of logical imitation. Whether corporations with high-quality or low-quality corporate governance, their EID is more positively influenced by corporations with high-quality corporate governance.

## 6. Conclusion and policy implications

This paper empirically examines the intra-industry peer effect of corporate EID using a panel fixed effects model based on data from 2015–2019 for Chinese heavily polluted listed corporations. The main findings to emerge from this study are as follows. There is an intra-industry peer effect in corporate EID; this effect exists in corporations of different ownership; social learning mechanism and environmental pressure mechanism are the channels to form the intra-industry peer effect of corporate EID; there are both intra-group imitation and inter-group imitation in the intra-industry peer effect of corporate EID.

This paper introduces the theory of peer effect in sociological research, verifies the intra-industry peer effect of corporate EID, and reveals the mutual influence of EID among corporations, thus providing a new perspective for the study of EID. Previous studies have only found the existence of this effect and have not further explored its causes. In this paper, based on information theory and competition theory, we found that the formation mechanism of intra-industry peer effect in corporate EID includes social learning and environmental protection pressure mechanisms. Corporations with inferior information or higher environmental protection pressure are more likely to be affected by industrial peer effects. The above research

**Table 9. Grouping regression results of corporate governance factors.**

| VARIABLES | (1) | (2) |
|---|---|---|
|  | **Sample of high-performance corporations** | **Sample of low-performance corporations** |
| Baedi | 0.178*** | 0.272*** |
|  | (3.45) | (5.27) |
| Saedi | 0.002 | -0.013 |
|  | (0.03) | (-0.21) |
| $Edi_{t-1}$ | 0.292*** | 0.252*** |
|  | (6.85) | (5.27) |
| Size | 2.189*** | 1.618** |
|  | (4.42) | (2.39) |
| Cr | -0.026 | -0.025 |
|  | (-1.51) | (-0.81) |
| Lev | -0.309 | 2.261 |
|  | (-0.27) | (1.60) |
| Roa | 8.317*** | -1.814 |
|  | (2.91) | (-0.78) |
| Tobin Q | -0.439*** | -0.610*** |
|  | (-3.05) | (-4.05) |
| Growth | 0.029 | -0.059 |
|  | (0.13) | (-0.36) |
| Big4 | -1.457*** | -3.410* |
|  | (-2.98) | (-1.90) |
| Opin | -0.062 | 0.864 |
|  | (-0.08) | (1.30) |
| Constant | -42.138*** | -29.009* |
|  | (-3.91) | (-1.94) |
| Observations | 931 | 879 |
| R-squared | 0.497 | 0.470 |
| Company FE | YES | YES |

Robust t-statistics in parentheses

*** $p < 0.01$,

** $p < 0.05$,

* $p < 0.1$.

results further enrich the theoretical system of EID. In addition, this paper also analyzes the impact path of this effect from the perspective of corporate characteristics. Corporate EID of intra-industry imitation path can be divided into parallel imitation (Large corporations imitate large corporations, high-performance corporations imitate high-performance corporations) and logical imitation (low-performance corporations imitate high-performance corporations). There is no situation of weak imitating weak (small corporations imitate small corporations, low-performance corporations imitate low-performance corporations). These interesting research findings can provide valuable references for policy formulation related to EID.

These findings have important implications for enacting policy related to promoting corporate EID. The government should make the most of the mutual imitation among corporations in the same industry to improve the overall EID level. It is necessary for the government to select corporations with a high level of EID as benchmark corporations in various industries and vigorously publicize and praise these benchmark corporations through the media.

Establishing benchmark corporations can provide a model of EID for corporations with inferior information. This way, these corporations can know what they need to disclose and how to disclose it. Meanwhile, applying appropriate pressure on environmental protection can promote active learning and imitation among corporations. Finally, the setting of benchmarking corporations needs to consider factors such as their scale and performance because corporations in the same industry tend to imitate strong performers.

Despite these meaningful results, limitations remain. First, this paper uses the indicator evaluation method to measure the indicator of EID. Even though the indicator design of this paper refers to the research results of many experts in this field, the subjective manual judgment makes the indicator evaluation method unable to evaluate the quality of corporate EID comprehensively. Future research is required to conduct an in-depth semantic analysis of corporate EID text based on machine learning to measure the level of EID more scientifically and accurately, which is beneficial to developing EID research. Secondly, this paper verifies the peer effect of corporate EID only from the perspective of the same industry. A further study could assess the peer effect of EID from the perspective of the same region, the same audit firm, the chain of shareholders, and the chain of directors. Finally, this study analyzed the peer effect's existence, formation mechanism, and transmission path but did not pay attention to the consequences of the peer effect. Considerably more work will need to be done to determine whether the EID peer effects lead corporations to make substantial environmental governance, such as environmental investment and green innovation.

## Supporting information

**S1 Data.**
(XLSX)

## Author Contributions

**Conceptualization:** Kewei Hu, Yugui Hao.

**Data curation:** Kewei Hu, Dan Yu.

**Formal analysis:** Kewei Hu.

**Funding acquisition:** Yugui Hao.

**Methodology:** Kewei Hu.

**Project administration:** Yugui Hao.

**Resources:** Kewei Hu.

**Supervision:** Dan Yu.

**Writing – original draft:** Kewei Hu.

**Writing – review & editing:** Kewei Hu, Dan Yu.

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
