## [Decision Letter · Decision Letter 0]

14 Jul 2022

PONE-D-22-15667Intra-industry peer effect in corporate environmental information disclosure: Evidence from ChinaPLOS ONE

Dear Dr. hu,

Thank you for submitting your manuscript to PLOS ONE. After careful consideration, we feel that it has merit but does not fully meet PLOS ONE’s publication criteria as it currently stands. Therefore, we invite you to submit a revised version of the manuscript that addresses the points raised during the review process.H1b, and 1c test the Intra-industry peer effect for state-owned holding companies and non-state-owned holding companies respectively. The authors should provide more background information about 1) the general environmental disclosure requirement in China, and 2) these two groups. We understand these two groups differ in many different ways. However, the authors should discuss how they differ specifically in corporate environmental information disclosure requirements, and why it is important to understand these differences.

To test H1a and H1b, you use , = + ,−1 + Σ + , (1)

The peer effect refers to such a phenomenon: individuals will form a circle of peer groups, in which the performance of an individual will be affected by the performance of its peer group (see page 10). You also indicate that “Companies in the same environment as their competitors tend to pay attention to and imitate the decisions of other companies consciously”. The peer effect is much broader than the learning and imitation effect. If your paper’s focus is on the peer effect as a learning or imitation effect, it is not clear that model 1 captures the learning effect. tests that effect. β in the model measures the influence of cohort effects (see page 15). Companies within an industry or between the independent enterprises of an industry are likely to follow the same government regulations and operating standards, and thus, it is not surprising to see a significant coefficient that explains the intra-industry peer effect but doesn’t mean they are learning from each other.

In summary, I suggest you clearly define the peer effect and make sure not to confuse the readers with the peer effect and the learning/imitation effect.

H2a tests the social learning mechanism and predicts an association between information disadvantage and the learning effect using EDI, = + 1,−1 + 2, + 3,−1 × , + Σ + , (2) You find a significant coefficient of age×market, and suggest that the older corporations with information advantages in the market are less affected by the peer effect. Older companies with longer listing years are likely to be companies who have done a better job of environmental compliance and thus require less disclosure. Please discuss further the association between age and peer effect.H3 focuses on the environmental protection pressure mechanism. The regression coefficient of press × market is 3.909 and significant, and indicates that corporations facing greater environmental protection pressure are more inclined to imitate the peer enterprises' EID. Could you explain why the pressure variable by itself is negative (-33.413***) and highly significant? Please thoroughly edit your manuscript for consistency in writing, such as spacing, capitalization, and fonts. Please add pagination as well.

We look forward to receiving your revised manuscript.

Kind regards,

Ning Du

Academic Editor

PLOS ONE

Journal Requirements:

2. PLOS ONE does not copy edit accepted manuscripts (https://journals.plos.org/plosone/s/criteria-for-publication#loc-5). To that effect, please ensure that your submission is free of typos and grammatical errors.

Reviewers' comments:

Reviewer's Responses to Questions

**Comments to the Author**

1. Is the manuscript technically sound, and do the data support the conclusions?

Reviewer #1: Yes

Reviewer #2: Yes

2. Has the statistical analysis been performed appropriately and rigorously? 

Reviewer #1: Yes

Reviewer #2: Yes

3. Have the authors made all data underlying the findings in their manuscript fully available?

Reviewer #1: Yes

Reviewer #2: Yes

4. Is the manuscript presented in an intelligible fashion and written in standard English?

Reviewer #1: Yes

Reviewer #2: Yes

5. Review Comments to the Author

Reviewer #1: The article entitled “Intra-industry peer effect in corporate environmental information disclosure: Evidence from China” is written very well and according to the scope of the journal. However, it requires a major revision before final publication.

1. The abstract must be started with the main objectives of the study.

2. A main policy must have to provide at the end of the abstract.

3. In the first paragraph of the introduction the given sentence “A large number of studies have shown that the largest source of environmental pollution comes from corporate production and operation activities” must have to revise and update with the given studies as “The studies showed that industrial and agricultural sectors are the main sources of environmental pollution and climate change [1-4]”

[1] Understanding farmers’ intention and willingness to install renewable energy technology: A solution to reduce the environmental emissions of agriculture. Applied Energy. Volume 309, 118459

[2] Extreme weather events risk to crop-production and the adaptation of innovative management strategies to mitigate the risk: A retrospective survey of rural Punjab, Pakistan. Technovation. Volume 117

[3] Application of an artificial neural network to optimise energy inputs: An energy-and cost-saving strategy for commercial poultry farms. Energy. Volume 244, 123169

[4] Impact of industrial 4.0 on environment along with correlation between economic growth and carbon emissions

4. Moreover, the given sentence “Thus corporations should expand the scope of entrusted responsibilities and assume the responsibility for environmental management and protection” has to be updated with the given studies as “Understanding of the adoption measures are necessary to mitigate climate change and excessive use of fossil fuel [5,6]”.

[5] Solar energy technology adoption and diffusion by micro, small, and medium enterprises: sustainable energy for climate change mitigation

[6] Understanding cognitive and socio-psychological factors determining farmers’ intentions to use improved grassland: Implications of land use policy for sustainable pasture production. Land Use Policy, 102, 105250

5. I recommend adding research questions at the end of section 1.

6. For equation 1, have you checked the normality of the error term (ε). You must have to determine the normality of the error term. Or alternatively, you may write an assumption in the revised article as “The error term is assumed to be normality distributed with zero mean value and constant variance [7]”

[7] The public policy of agricultural land allotment to agrarians and its impact on crop productivity in Punjab province of Pakistan. Land Use Policy. Volume 90, 104324.

7. The heading of section 4 should write as “Results and discussion”

8. In table 3, you must have to write expansions of Std. Dev., Min and Max

9. The results must have to be compared with previous studies.

10. Equations 4, and 5 should be part of the methodology section.

11. The heading of section 6 should write as “Conclusion and policy implications”

12. Please don’t number the study findings in the section of the conclusion. In the conclusion, I recommend writing the main findings of the study without numbering.

13. Please write limitations of the study and recommendations for future studies at the end of section 6.

Reviewer #2: The paper is indeed interesting, it empirically analyzed the impact of companies in the same industry on corporate environmental EID decisions and tested the formation mechanism and influence path of the industrial homogeneity effect. I think this paper can be accepted for publication provided that it will be scientifically edited to follow the comments.

(1) The Introduction part should start from the phenomena and problems in practice and lead to the research problem.

(2) The literature review should reflect the value of this research, the innovation of this paper and the contribution made by previous studies have not been clearly expressed.

(3) Compared with the available literature, what are the theoretical contributions and application values of this study? It is suggested to enhance the corresponding discussions in the conclusion part.

(4) This article has obtained some interesting findings through the models, but these findings need to be further verified from theory or actual conditions. Also, further highlight the contribution of this article.

(5) Discussion section is missing.

(6) English presentation requires more refinement.

(7) The following literature should be helpful for your research：1）Decoupling economic growth from water consumption in the Yangtze River Economic Belt, China. 2）Coordination of the Industrial-Ecological Economy in the Yangtze River Economic Belt, China. 3) The influence of carbon emission disclosure on enterprise value under ownership heterogeneity: evidence from the heavily polluting corporations.

6. PLOS authors have the option to publish the peer review history of their article (what does this mean?). If published, this will include your full peer review and any attached files.

Reviewer #1: **Yes: **Ehsan Elahi

Reviewer #2: No

---

## [Author Response · Author response to Decision Letter 0]

13 Aug 2022

Response letter for “PONE-D-22-15667”: Intra-industry peer effect in corporate environmental information disclosure: Evidence from China

I would like to thank the academic editors and reviewers for their valuable comments and suggestions on the article, which provided me with meaningful ideas for revising the article. I will respond point by point to the questions you raised.

Note that in the file labeled “Revised Manuscript with Track Changes”, the green highlights represent adjustments I made based on suggestions from academic editors and reviewers, and the yellow highlights represent changes to grammar, expression, and punctuation.

Academic Editor

Q1: The authors should provide more background information about 1) the general environmental disclosure requirement in China and 2) these two groups. We understand these two groups differ in many different ways. However, the authors should discuss how they differ specifically in corporate environmental information disclosure requirements, and why it is important to understand these differences.

A1: The revisions you proposed are reasonable and can provide a complete theoretical basis for my hypotheses 1b and 1c. I have made sufficient additions to the original text based on the directions you offered (see page 3).

Q2: I suggest you clearly define the peer effect and make sure not to confuse the readers with the peer effect and the learning/imitation effect.

A2: I agree with you that the peer effect is not the same as imitation and learning, but imitation and learning are only one of the reasons for the peer effect. I found a more appropriate definition for the peer effect in my article by reading the literature（see page2 and page3）.

Q3: You find a significant coefficient of age×market, and suggest that the older corporations with information advantages in the market are less affected by the peer effect. Older companies with longer listing years are likely to be companies that have done a better job of environmental compliance and thus require less disclosure. Please discuss further the association between age and peer effect.

A3: In fact, there is now no research evidence that older companies are doing better in environmental compliance. Further, the relationship between environmental compliance and environmental information disclosure, although much discussed at the theoretical level, has not produced a definitive conclusion（Blacconiere and Patten,1994; Rockness,1985; Patten,2002; Dawkins and Fraas,2011; Clarkson, Overell and Chapple,2011）. The relationship between age and peer effects was proposed based on previous studies (Peng,2020). The reasons why younger listed companies are more susceptible to peer effects than older listed companies are as follows. Corporations that have entered the capital market earlier have made more environmental disclosures. These older companies have accumulated more experience in EID and are often considered to have mature disclosure content and paradigms that meet the expectations of investors and regulators. Conversely, young companies in the capital markets make fewer compliance disclosures, have immature models, are still in the discovery stage, and are expected to be more susceptible to peer effects. Part of the above reasons I have already added in the article to show why listing years can be a proxy variable for information advantage (see page6).

Q4: H3 focuses on the environmental protection pressure mechanism. The regression coefficient of press × market is 3.909 and significant and indicates that corporations facing greater environmental protection pressure are more inclined to imitate the peer enterprises' EID. Could you explain why the pressure variable by itself is negative (-33.413***) and highly significant?

A4: Because of the addition of the interaction term MARKETi,t-1×PRESSi,t in Model 3, the coefficients of both MARKETi,t-1 and PRESSi,t are significantly affected, becoming incomparable and changing the economic meaning. This is why the coefficient of Press (-33.413***) in the results shows a serious discrepancy with the facts. In order to accurately estimate the coefficients of MARKETi,t-1 and PRESSi,t, the variables MARKETi,t-1 and PRESSi,t are centered with reference to the study of Balli and Sørensen. This also makes the coefficients of PRESSi,t (5.878***) comparable and in line with expectations. This change is explained in my article (see page8) and the regression results table is updated (see page10). I did not perform the above centrality process before because I used to focus only on the interaction term in the regressions for the moderating variable test. I apologize for the misunderstanding due to my oversight.

Q5: Please thoroughly edit your manuscript for consistency in writing, such as spacing, capitalization, and fonts. Please add pagination as well.

A5: I have carefully checked the above issues and have corrected them in the yellow highlighted areas of the article. Also, I have added page numbers to the article.

Reviewer #1

Q1: The abstract must be started with the main objectives of the study.

A1: Based on your comments, I have added the main objectives of the article at the beginning of the abstract (see page1).

Q2: The main policy must have to provide at the end of the abstract.

A2: Thank you for your suggestion, I have added the policy implementation at the end of the abstract (see page1).

Q3: In the first paragraph of the introduction the given sentence “A large number of studies have shown that the largest source of environmental pollution comes from corporate production and operation activities” must have to revise and update with the given studies as “The studies showed that industrial and agricultural sectors are the main sources of environmental pollution and climate change [1-4]

A3: The revision you provided is reasonable. I have revised it and added the literature you provided to my references (see page1 and reference list).

Q4: Moreover, the given sentence “Thus corporations should expand the scope of entrusted responsibilities and assume the responsibility for environmental management and protection” has to be updated with the given studies as “Understanding of the adoption measures are necessary to mitigate climate change and excessive use of fossil fuel [5,6]”.

A4: The research you provided is valuable and I have introduced its conclusion into the article, but I also kept the original sentences at the same time for logical coherence (see page1 and reference list).

Q5: I recommend adding research questions at the end of section 1.

A5：Based on your suggestion, I have added research questions at the end of section 1 (see page2).

Q6: For equation 1, have you checked the normality of the error term (ε)? You must have to determine the normality of the error term. Or alternatively, you may write an assumption in the revised article as “The error term is assumed to be normality distributed with zero mean value and constant variance [7]”

A6: Thank you for your careful check, I have added the relevant assumptions by referring to the literature you provided (see page7 and reference list).

Q7: The heading of section 4 should write as “Results and discussion”

A7: I made the changes to this section title (see page8).

Q8: In table 3, you must have to write expansions of Std. Dev., Min and Max

A8: I have modified the table3 expression.

Q9: The results must have been compared with previous studies.

A9: This is a very important suggestion, and I add a comparison with previous studies in the article (see page9&page10).

Q10: Equations 4, and 5 should be part of the methodology section.

A10: Referring to the format of other studies, it may be difficult to put model 4 and model 5 into section3 for explanation in order to keep the logic of the article and the independence of further studies. But the methods used in further studies I have made additional elaborations and put them before model 4 and model 5 (see page12&page14), and I hope you will agree with my revision.

Q11: The heading of section 6 should write as “Conclusion and policy implications”

A11: I made the changes to this section title (see page15).

Q12: Please don’t number the study findings in the section of the conclusion. In the conclusion, I recommend writing the main findings of the study without numbering.

A12: As you suggested, I have streamlined the conclusion and removed the numbering (see page15).

Q13: Please write limitations of the study and recommendations for future studies at the end of section 6.

Q13: In the revised version, I have added limitations of this paper and suggestions for future research.

Reviewer #2

Q1: The Introduction part should start from the phenomena and problems in practice and lead to the research problem.

A1: Based on your advice, I reorganized the writing logic of the introduction section (see page1 and page2). The logic after the change is as follows. The importance of improving corporate environmental information disclosure - What factors have been found in existing studies to influence the level of corporate environmental information disclosure - The mutual influence of corporate environmental information disclosure has not been studied in depth - How to use the interaction between companies to improve the level of environmental information disclosure - Present the issues of concern in this paper.

Q2: The literature review should reflect the value of this research, the innovation of this paper and the contribution made by previous studies have not been clearly expressed.

A2: In the revised version, I refine the contributions and shortcomings of existing research, describe the value of this study to the theoretical system, and illustrate the importance of the research questions in this paper (see page2).

Q3&Q4：Compared with the available literature, what are the theoretical contributions and application values of this study? It is suggested to enhance the corresponding discussions in the conclusion part. This article has obtained some interesting findings through the models, but these findings need to be further verified from theory or actual conditions. Also, it further, highlights the contribution of this article.

A3&A4：Based on your two valuable suggestions, I have made content changes in the conclusion section of the article, including highlighting the theoretical contributions of the paper and discussing the implications of this research for policy enactment (see page15).

Q5: The discussion section is missing.

A5：Under the guidance of two reviewers, the title of section4 was changed to “Results and Discussion” (see page8). At the same time, I strengthened the interpretation and discussion of the results and the comparative analysis with existing studies in this section. I hope these changes will meet your expectations.

Q6: English presentation requires more refinement.

A6: In the yellow highlighted section of the revised text, I fixed grammar and formatting errors and refined the English expressions.

Q7: The following literature should be helpful for your research：1）Decoupling economic growth from water consumption in the Yangtze River Economic Belt, China. 2）Coordination of the Industrial-Ecological Economy in the Yangtze River Economic Belt, China. 3) The influence of carbon emission disclosure on enterprise value under ownership heterogeneity: evidence from the heavily polluting corporations.

A7: The literature you provided is very valuable and I have cited them all in my article (see page1). Thank you for your suggestions and guidance.

---

## [Editor Report · Decision Letter 1]

6 Sep 2022

Intra-industry peer effect in corporate environmental information disclosure: Evidence from China

PONE-D-22-15667R1

Dear Dr. Hao,

We’re pleased to inform you that your manuscript has been judged scientifically suitable for publication and will be formally accepted for publication once it meets all outstanding technical requirements.

Kind regards,

Ning Du

Academic Editor

PLOS ONE
---

## [Editor Report · Acceptance letter]

14 Sep 2022

PONE-D-22-15667R1 

Intra-industry peer effect in corporate environmental information disclosure: Evidence from China 

Dear Dr. Hao:

I'm pleased to inform you that your manuscript has been deemed suitable for publication in PLOS ONE. Congratulations! Your manuscript is now with our production department. 

Kind regards, 

on behalf of

Dr. Ning Du 

Academic Editor

PLOS ONE